# When Can Model-Free Reinforcement Learning be Enough for Thinking?

**Josiah P. Hanna**
Computer Sciences Department
University of Wisconsin – Madison
jphanna@cs.wisc.edu

**Nicholas E. Corrado**
Computer Sciences Department
University of Wisconsin – Madison
ncorrado@wisc.edu

## Abstract

Recent work on large language models has demonstrated the use of model-free reinforcement learning (RL) to train reasoning-like capabilities. The emergence of "thinking" through model-free RL is interesting as thinking actions neither produce reward nor change the external world state to one where the agent is more likely to get reward. This paper seeks to build a domain-independent understanding of when model-free RL will lead to such "thinking" as a strategy for reward maximization. To build this understanding, we first introduce a theoretical model which we call a *thought Markov decision process* (MDP). Thought MDPs minimally extend the classical MDP model to include an abstract notion of thought state and thought action. Using the thought MDP model, we prove the importance of policy initialization in determining whether or not thinking emerges and show formally that thought actions are equivalent to the agent choosing to perform a step of policy improvement before continuing to act. We then show that open-source LLMs satisfy the conditions that our theory predicts are necessary for model-free RL to produce thinking-like behavior. Finally, we hypothesize sufficient conditions that would enable thinking to be learned outside of language generation and introduce a toy domain where a combination of multi-task pre-training and designated thought actions enable more data-efficient RL compared to non-thinking agents.

## 1 Introduction

Dual process theory divides cognitive processing into System 1 and System 2 processing [Wason and Evans, 1974, Kahneman, 2003]. System 1 processing is fast, effortless, but potentially imprecise, whereas System 2 is slow, effortful, but potentially produces better decisions. In the field of computational reinforcement learning (RL), model-free learning is reminiscent of System 1 behavior, whereas model-based decision-time planning is reminiscent of System 2. While dual process theory is a popular model for understanding human cognition [Kahneman, 2011], there is no single clear candidate architecture for its duplication in artificial agents.

In the context of large language models (LLMs), recent work by Guo et al. proposed a unified model whereby System 2 type "thinking" emerged as a consequence of model-free RL applied to solve mathematics problems (2025). While this "thinking" appears to resemble thoughts that a human would have when answering a given query, these outputs arise solely as subservient to reward maximization. The approach is interesting as it suggests that a form of System 2 processing can emerge when we view thinking as control and reinforce thought patterns that lead to reward. Our objective is to develop a domain-independent understanding of the conditions under which model-free RL will select for thinking behavior. Specifically, we want to answer the question:

*Under what conditions will model-free reinforcement learning give rise to thinking as a strategy for reward maximization?*

39th Conference on Neural Information Processing Systems (NeurIPS 2025).

Informally, we define thinking to be actions that do not directly produce reward or affect the external state of an agent's environment but that lead the agent to take a course of action that increases the reward it will receive in the future. To answer our research question, we first formulate a minimal extension of the classical MDP model to explicitly model thought actions and a notion of a controllable thought state. We then show how policy initialization plays a central role in whether policy iteration will select thought actions. Under our theoretical model, we will show that thinking can be viewed as selecting between a set of sub-policies that are already contained in the learning agent's policy function and that thought actions can be interpreted as the agent choosing to run one or more steps of policy improvement before continuing to act. We then discuss how LLMs instantiate our thought MDP formalism and provide empirical evidence that they exhibit the necessary conditions for thinking to emerge. Our final contribution is to introduce a simple domain and multi-task pre-training set-up that induces the condition under which model-free RL will discover thinking behavior. This simple domain provides a basis for future work studying agents that learn to think and act. We conclude by discussing open questions and directions for future work raised by this model of deliberative thinking in RL.

## 2 Related Work

In this section, we discuss the prior literature related to System 2 processing in RL as well as prior models of "mental" actions an agent can take.

**Decision-time Planning** A hallmark of System 2 processing is deliberative planning as opposed to reflexive action. Thus, a natural way to instantiate System 2 in artificial agents is to use decision-time planning, e.g., Monte Carlo tree search [Kocsis and Szepesvári, 2006], and many works in the RL literature have used this approach [e.g., Anthony et al., 2017, Silver et al., 2016, 2017, Silver, 2009, Schrittwieser et al., 2021, Shah et al., 2020]. Decision-time planning is particularly effective when the agent's policy or value function is sub-optimal but the agent possesses an accurate model of state transitions [Sutton and Barto, 2018]. In this case, planning can be viewed as focused policy improvement to improve the decision at the agent's current state before it acts. As we will show, the idea of thinking as local policy improvement also applies when an RL agent learns to think in our new thought MDP model. The key difference is that we consider a more abstract model of thinking without any notion of forward prediction and search. Furthermore, there is also no natural answer to the question of when to stop planning and act with most decision-time planning methods. MCTS is an anytime algorithm and the general practice is to allow as much search is possible in the application domain [Silver et al., 2016, 2017]. On the other hand, model-free learning in our thought MDP model will directly relate the decision about when to stop thinking to the objective of return maximization. Some work in the (non-learning) planning literature has also studied the question of when to think vs. when to act [Cashmore et al., 2019, Coles et al., 2024].

**Learning to Plan** As neural networks are general-purpose function approximators, they can, in principle, learn algorithmic planning or reasoning computations and prior work has attempted to induce planning capabilities through specialized architectures [Tamar et al., 2017, Farquhar et al., 2018, Guez et al., 2019, 2018, Niu et al., 2018, Weber et al., 2018, Sykora et al., 2020, Schleich et al., 2019, Oh et al., 2017]. Guez et al. showed that planning-like computations could emerge through model-free learning (2019) and Bush et al. [2025] applied mechanistic interpretability approaches to show that the approach of Guez et al. [2019] did in fact learn planning computations (i.e., the network internally learned to propose and evaluate future candidate action sequences). While such works also show that model-free RL can produce thinking behavior, the main difference is that thinking refers to the computation done by the agent's policy, whereas we study thinking as an action selected by the agent's policy. More similar to this conception of thinking, the thinker architecture (Chung et al. [2023], Wang et al. [2025]) trains a policy to select actions both for environment interaction and planning in a learned model. In contrast, we consider a model-free conception of thinking.

**Reasoning in LLMs** In the past couple of years, a large number of methods have been developed to create System 2 capabilities in LLMs, with particular emphasis on math and coding problems. One prominent approach is chain-of-thought (CoT) prompting in which a user changes their query to include explicit examples of correct responses [Wei et al., 2023]. A more basic variant of CoT is just to prompt the model to "think step by step" [Kojima et al., 2023]. Many other variations have also been proposed [e.g., Yao et al., 2023a,b, Wang et al., 2023]. Another paradigm has been to augment

output generation with an explicit search procedure [e.g., Khanov et al., 2024, Liu et al., 2023, Zhou et al., 2023, Zhang et al., 2023, Chen et al., 2024]. Finally, recent works have used model-free RL to train LLMs to output CoT reasoning [Guo et al., 2025, OpenAI et al., 2024]. Our work takes inspiration from these works but aims to go beyond LLMs in understanding when thinking-like behavior can emerge in RL agents.

**Cognition as Action**    The idea of thinking as a form of action has a long history in AI and RL. Minsky theorized on the mind as a society of agents whose collective actions produce cognition and action [Minsky, 1986]. Klopf's hedonistic neuron hypothesis modelled neurons as individual RL agents [Klopf, 1982]. This hypothesis has led to a line of work studying alternative neural network architectures in which artificial neural activity is produced by the stochastic policies of simple RL agents [Thomas, 2011, Kostas et al., 2019, Gupta et al., 2021]. More similar to our work, some prior work has considered augmenting the agent's environment action-space with a form of mental action. These works have focused on the challenge of memory in partially observable domains and using explicit memory read or write actions as an alternative to recurrent neural networks. For instance, Peshkin et al. [2001] augment an RL agent with a set of memory write actions and the contents of the memory are then provided to the agent as an additional input along with the environment state; Zhang et al. [2015] extend this approach to robot manipulation tasks. Various neural architectures have been developed with external memory that can be written to and read from [e.g., Graves et al., 2014, Zaremba and Sutskever, 2016] and Oh et al. [2016] explored these architectures for RL. The thought MDP model differs in that the agent has access to a Markov state representation and so thought actions serve to manipulate the agent's policy as opposed to remembering details of the past. Prior work has used the framework of rational meta-reasoning to study computation selection as a form of action [Hay et al., 2012] and even consider using RL to learn a computation selection policy [Callaway et al., 2018]. These works differ in that they study choosing among concrete computations as opposed to the abstract model of thinking as manipulating the agent's internal state that we consider.

**The Options Framework**    Finally, thought MDPs are related to the options framework that is often used in hierarchical RL [Sutton et al., 1999]. In the options framework, the agent's policy is over a set of options where options are either primitive actions or sub-policies. The crucial difference is that in the options framework, the selection of an option and the execution of that option's first action both occur within a single time step, whereas this execution would take at least two steps in a thought MDP. This difference suggests that thought MDPs might naturally model the cost of switching options, particularly when the agent cannot directly set its thought state and must instead think for multiple steps to select the best option. Thus, thought MDPs might be particularly useful as an alternative to the options with deliberation cost model [Harb et al., 2018].

## 3   Formal Model

In this section, we first formalize the standard RL problem using the MDP formalism and then introduce the thought MDP model to explicitly model thinking.

### 3.1   RL in Markov Decision Processes

RL environments are typically modeled as Markov decision processes (MDPs). Formally, an MDP is a tuple, $\langle \mathcal{S}, \mathcal{A}, p, r, \gamma \rangle$, where $\mathcal{S}$ is the environment state space, $\mathcal{A}$ is the set of actions that the agent can take to influence the environment state, $p : \mathcal{S} \times \mathcal{A} \to \Delta(\mathcal{S})$ is a stochastic state transition function, $r : \mathcal{S} \times \mathcal{A} \to \mathbb{R}$ is a scalar-valued reward function, and $\gamma$ is a discount factor. At any moment in time, the agent is in state $S_t$, takes an action, $A_t$, receives a reward, $R_t = r(S_t, A_t)$, and transitions to a new state, $S_{t+1}$. Then the process repeats from $S_{t+1}$ until a special terminal state, $s_\infty$, is reached. The agent selects actions according to a policy, $\pi : \mathcal{S} \to \Delta(\mathcal{A})$. The value of using a policy from a particular state, $s$, is defined to be $v_\pi(s) \coloneqq \mathbf{E}_\pi[\sum_{t=0}^\infty \gamma^t R_t | S_0 = s]$. In RL, the agent's objective is to find a policy that maximizes $v_\pi(s)$ in all states.

### 3.2   Thought MDPs

We now extend the MDP model to explicitly model thinking. We formally define a *thought MDP* as the tuple $\langle \mathcal{S}, \mathcal{A}, p, r, \gamma, \mathcal{T}, \mathcal{C}, p_\mathcal{T} \rangle$, where $\mathcal{S}$ and $\mathcal{A}$ are the environment's state and action space respectively and $p, r, \gamma$ are defined as they are for MDPs. We add $\mathcal{T}$ as the set of *thought states*, $\mathcal{C}$ as

the set of *thought actions*, and $p_{\mathcal{T}} : \mathcal{S} \times \mathcal{T} \times \mathcal{C} \to \Delta(\mathcal{T})$ as the *thought transition function*. Recalling our informal definition of thinking in the introduction, we emphasize that thought states and actions do not affect environment state transitions and rewards. The agent's objective remains to maximize cumulative discounted reward across all environment states.[1]

**Policies in Thought MDPs**  There are different ways to define the agent's policy in a thought MDP. In this work, we formalize policies as a mapping $\pi : \mathcal{S} \times \mathcal{T} \to \Delta(\mathcal{A} \cup \mathcal{C})$. This choice means that the agent can select either an environment action or a thought action at any interaction time-step but not both. We make this choice to connect to work in LLMs that inspired this model, but an alternative is that the policy is a mapping $\pi : \mathcal{S} \times \mathcal{T} \to \Delta(\mathcal{A} \times \mathcal{C})$, i.e., the agent can think and act at the same time. Such modelling might be more appropriate for real-time domains where the agent cannot simply sit and think as the rest of the world evolves around it. There, of course, may be other alternatives that could be considered in future work. Below, we will use the notation $\pi(\tau)$ to refer to the agent's state-dependent policy with the thought state fixed at $\tau$.

**Interaction in Thought MDPs**  In a thought MDP, interaction proceeds as follows. Episodes begin in an environment state, $S_0$, and thought state, $\tau_0$, and the agent chooses either an environment action or thought action according to its policy. Let $A_0$ be the random variable denoting this choice. If $A_0 \in \mathcal{A}$ then the environment state, $S_1$, at the next time-step is sampled from $p(\cdot|S_0, A_0)$, the thought state $\tau_1$ keeps the value of $\tau_0$, and the agent receives reward $R_0 := r(S_0, A_0)$. Conversely, if $A_0 \in \mathcal{C}$, then $S_1$ keeps the value of $S_0$, $\tau_t$ is sampled from $p_{\mathcal{T}}(\cdot|S_0, \tau_0, C_0)$, and the agent receives $R_0 = 0$. Then the process repeats with the agent selecting either an environment or thought action from its policy until the agent reaches a special terminal state $s_\infty \in \mathcal{S}$. For thought MDPs, we define the value function to be $v_\pi(s, \tau) := \mathbf{E}_\pi[\sum_{t=0}^\infty \gamma^t R_t | S_0 = s, \tau_0 = \tau]$.

**Key Assumptions**  We aim to introduce a general model of thinking in MDPs, however, our subsequent analysis will make two assumptions which we state formally here. First, we assume that thought state transitions are deterministic in order to simplify our analysis and because it is also the case for LLM agents.

**Assumption 1** (Deterministic Thought Transitions). $\forall s \in \mathcal{S}, \tau \in \mathcal{T}, c \in \mathcal{C} \; p_{\mathcal{T}}(\tau'|s, \tau, c) = 1$ for one and only one $\tau' \in \mathcal{T}$. We will write $p_{\mathcal{T}}(s, \tau, c) \in \mathcal{T}$ to denote the thought state that results from taking $c$ in $(s, \tau)$.

Second, we assume that all rewards are non-negative as otherwise thinking could emerge as a strategy solely for the purpose of putting off receiving a negative reward, i.e., if all rewards are negative then the agent will be incentivized to just keep taking thought actions rather than environment actions.

**Assumption 2** (Non-negative Rewards). $\forall s \in \mathcal{S}, a \in \mathcal{A}, r(s, a) \geq 0$.

Finally, we assume reachable positive reward from all states, as otherwise there will exist states where the agent is indifferent to whether it takes a thought action or an environment action.

**Assumption 3** (Reachable Positive Reward). $\exists s \in \mathcal{S}, a \in \mathcal{A}$ with $r(s, a) > 0$ and $\forall \tilde{s} \in \mathcal{S}$ there exists a policy such that the probability of transitioning from $\tilde{s}$ to $s$ in a finite number of steps is greater than zero.

**Modeling of Time**  Our formalism raises questions about how time should be treated for the two forms of action. First, should the discount factor be applied equally for both thinking and non-thinking time-steps? Equal application discourages thinking as thought actions do not influence reward either directly or indirectly through the environment state. Nevertheless, as we shall see, thought actions still might be selected if they ultimately cause the agent to choose a better environment action. Alternatively, we could apply a different discount factor for thinking time-steps to reflect the actual time-delay of thinking compared to acting in a given domain. For example, if thinking takes $(1/k)$ the time of any environment action then we could use a discount of $\gamma^{\frac{1}{k}}$. In this work, we will assume that the discount is applied the same at both thinking and non-thinking time steps. The second related issue is that the proposed model assumes that the environment state remains constant while the agent takes thought actions. Such an assumption might be reasonable for relatively static environments (such as

---

[1]We base the thought MDP model on the widely used discounted problem formulation. In Appendix Appendix B, we extend the thought MDP model and theoretical results to the undiscounted, fixed-horizon setting which more closely resembles contemporary applications of RL to LLMs.

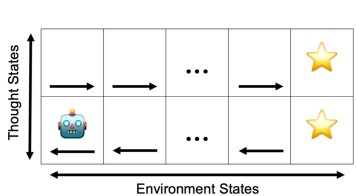
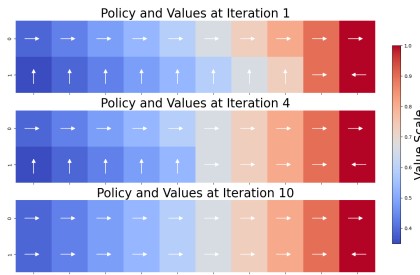

Figure 1: (left) An example thought MDP with $|\mathcal{T}| = 2$. We use $|\mathcal{S}| = 10$ in our illustrative results. The agent receives a reward when it reaches the goal environment state on the far right. The agent can move left or right in the environment state space and up and down in the thought state space. We use $\gamma = 0.9$ for both thinking and non-thinking time-steps. (right) Evolution of the policy and state values for 1, 4, and 10 iterations of policy iteration. The policy is initialized as shown on the left. Colors indicate value and arrows indicate the action that the policy would take.

generating text or board games) but is problematic in dynamic environments (such as autonomous driving) where waiting to act can be consequential. Again, for simplicity, we will focus on the static environment case, but note this issue as another interesting direction to refine the model.

**Optimality**  Thought MDPs are MDPs with state space $\mathcal{S} \times \mathcal{T}$ and action space $\mathcal{A} \cup \mathcal{C}$ and consequently there will always be at least one deterministic optimal policy [Sutton and Barto, 2018]. Furthermore, we can show that this optimal policy will never select thought actions.

**Proposition 4.** *Any optimal policy, $\pi^\star$, for a thought MDP does not take thought actions: $\pi^\star(s, \tau) \in \mathcal{A}, \forall s \in \mathcal{S}, \tau \in \mathcal{T}$.*

*Proof.* The proof is by contradiction. Suppose $\pi^\star$ is an optimal policy and $\exists s \in \mathcal{S}, \tau \in \mathcal{T}$ such that $\pi(s, \tau) \in \mathcal{C}$. Because thought actions cannot produce reward, the optimal policy must eventually reach a thought state, $\widetilde{\tau}$, where $\pi^\star(s, \widetilde{\tau}) \in \mathcal{A}$. If that happens after $k$ thought actions, then $v_{\pi^\star}(s, \tau) = \gamma^k v_{\pi^\star}(s, \widetilde{\tau})$. But then the policy could be strictly improved by setting $\pi^\star(s, \tau) \leftarrow \pi^\star(s, \widetilde{\tau})$. This contradicts the assumption that $\pi^\star$ was optimal and completes the proof. $\qquad\square$

## 4   Policy Initialization Determines Emergence of Thinking

If the optimal policy would never take a thought action, why should we expect thinking to emerge as a strategy during policy improvement? In this section, we begin to answer this question by showing the key role that policy initialization plays in determining the emergence of thinking. In particular, we first consider exact policy iteration within a thought MDP and provide an illustrative example and a formal result showing how thinking can emerge. We use policy iteration for analysis as essentially all model-free RL algorithms can be understood as instances of generalized policy iteration [Sutton and Barto, 2018]. We then provide a second formal result showing that thought actions can reduce the effective horizon [Laidlaw et al., 2023] for the special case of goal-MDPs.

**An illustrative example**  For exposition's sake, suppose $\mathcal{T} = \{\tau_0, \tau_1\}$, in which case we can view the agent's policy, $\pi$, as consisting of two sub-policies, $\pi(\tau_0)$ and $\pi(\tau_1)$. Now consider the example depicted in Figure 1 where $\pi(\tau_0)$ is initialized sub-optimally (always takes an action that leads away from the goal) while $\pi(\tau_1)$ is initialized to be optimal. We show the progress of policy iteration in this domain in Figure 1. After the first iteration of policy improvement, the policy has learned to first change the agent's thought state to $\tau_1$ (top row) from which it then follows $\pi(\tau_1)$ to reach the goal. After four iterations, in environment states close to the goal, the policy just directly moves right without changing its thought state while continuing to first take a thought action in states that are far from the goal. Finally, after ten iterations, the policy converges to the optimal policy and simply moves to the right without thinking. This example shows that, while thinking is sub-optimal in the long-run, it can be beneficial early in learning by allowing the agent to use sub-policies already contained in its policy function.

To formally understand how policy initialization determines the emergence of thinking, we analyze the policy improvement step of policy iteration.

**Theorem 5.** *Let $\pi$ be a policy in a thought MDP such that $\pi(s, \tau) \in \mathcal{A}$ for some environment state $s$ and thought state $\tau$. If the policy improvement step of policy iteration sets $\pi'(s, \tau) \leftarrow c$ for $c \in \mathcal{C}$, then $v_\pi(s, \tau') > v_\pi(s, \tau)$, where $\tau'$ is the thought state resulting from taking $c$ in $(s, \tau)$.*

*Proof.* Policy iteration sets $\pi'(s, \tau) \leftarrow \arg\max_{a \in \mathcal{A} \cup \mathcal{C}} q_\pi(s, \tau, a)$ where

$$q_\pi(s, \tau, a) := \begin{cases} r(s, a) + \gamma \mathbf{E}_{s'}[v_\pi(s', \tau)] & \text{if } a \in \mathcal{A} \\ \gamma v_\pi(s, \tau') & \text{if } a \in \mathcal{C}. \end{cases}$$

Since the policy improvement step is selecting a thought action, we have that:

$$\forall a \in \mathcal{A}, q_\pi(s, \tau, a) < \gamma v_\pi(s, \tau').$$

Since $\pi(s, \tau)$ was in $\mathcal{A}$ before the update we have that:

$$v_\pi(s, \tau) = q_\pi(s, \tau, \pi(s, \tau)) < \gamma v_\pi(s, \tau') < v_\pi(s, \tau') = q_\pi(s, \tau', \pi(s, \tau')).$$

Thus, a thought action is selected when the policy is such that changing the thought state to $\tau'$ will lead to a greater expected return from $s$ than when leaving the thought state at $\tau$. $\square$

Although, Theorem 5 does not directly say anything about policy initialization, the condition for a thought action to be selected requires $\pi(\tau)$ and $\pi(\tau')$ to be somehow set up as different such that there could be an advantage in shifting from $\tau$ to $\tau'$. Thus, policy initialization is critical for whether thinking will emerge or not.

Theorem 5 deals with exact policy iteration and does not address the impact of policy initialization on the emergence of thinking when learning from samples. We next turn to the concept of the *effective horizon* introduced by Laidlaw et al. [2023]. The effective horizon was introduced as a problem complexity parameter for MDPs that aligns with the observed difficulty of common RL benchmarks. Laidlaw et al. then derived sample complexity bounds for model-free RL algorithms in terms of the effective horizon and showed that these bounds were predictive of the success of deep RL algorithms on benchmark problems. We will build upon their specific result on goal MDPs.

**Definition 6.** A goal (thought) MDP is an (thought) MDP with a set of absorbing environment states, $\mathcal{S}_{\texttt{goal}} \subset \mathcal{S}$, and $r(s, a) = 1$ if $s \in \mathcal{S}_{\texttt{goal}}$ and 0 otherwise.

For goal MDPs, Laidlaw et al. [2023] showed that the effective horizon, $H$ can be upper-bounded by $1 + \log_\mathcal{A} \frac{\log 2l}{p}$ where $l$ is a maximum episode length, $p \leq \Pr_{\texttt{expl}}(s_T \in \mathcal{S}_{\texttt{goal}}|s_t, a_t)$ is a lower bound on the probability of the initial exploration policy, $\pi_{\texttt{expl}}$, discovering a goal state after taking action $a$ in state $s$ at any time-step. Intuitively, if $p$ is larger, then an RL algorithm has an easier time discovering rewarding action sequences. We next show that thought actions can be used to reduce the effective horizon of a given instance. Note that Laidlaw et al. [2023] proved their results assuming deterministic MDPs, so we also adopt this assumption for the next proposition.

**Proposition 7.** *Suppose we have a goal thought MDP that has $\mathcal{T} = \{\tau_0, \tau_1\}$. Let the initial policy, $\pi$, be such that there is some $p_0$ such that $\Pr(s_l \in s_{\texttt{goal}}|s, a, \pi(\tau_0)) > p_0$ for all $(s, a) \in \mathcal{S} \times \mathcal{A}$ and some $p_1$ such that $\Pr(s_l \in s_{\texttt{goal}}|s, a, \pi(\tau_1)) > p_1$ for all $(s, a) \in \mathcal{S} \times \mathcal{A}$. Let $c_s$ be the thought action such that $\tau_1 = p_\mathcal{T}(s, \tau_0, c_s)$ and let $p_c$ lower bound the probability $\pi(c_s|s, \tau_0)$. If $p_c \cdot p_1 > p_0$, then thought actions reduce the upper bound on the effective horizon of the MDP.*

See Appendix A for the proof. Intuitively, policy $\pi(\tau_1)$ has a higher probability of finding a goal state from any state compared to simply running $\pi(\tau_0)$. If the probability of taking a thought action that changes $\tau_0$ to $\tau_1$ is not too low then it is easier to find a goal state by first changing the thought state and then executing $\pi(\tau_1)$. Again, this result suggests that the benefit of thinking depends crucially on policy initialization.

**Thinking as a Policy Improvement Operator**

One interpretation of Theorem 5 is that thought actions can function as policy improvement operators applied to a particular state. This interpretation is interesting as it aligns with the use of decision-time planning in RL to refine an agent's choice of action in a way that focuses computation on its current

state [Sutton and Barto, 2018]. While thought actions do not involve look-ahead search with a model, Theorem 5 shows that their utility is also in providing local policy improvement. Theorem 5 shows this utility for the choice of a single thought action and we also present a corollary showing that if policy improvement produces a policy that thinks for consecutive steps in $s$ then each thinking step will further improve upon the action $\pi(s, \tau)$.

**Corollary 8.** *Let $c$ be a thought action that leads from $\tau$ to $\tau'$ and $c'$ be a thought action that leads from $\tau'$ to $\tau''$. If, in some environment state $s$, the policy improvement step of policy iteration sets $\pi'(s, \tau) \leftarrow c$ and $\pi'(s, \tau') \leftarrow c'$, then $v_\pi(s, \tau'') > v_\pi(s, \tau') > v_\pi(s, \tau)$.*

*Proof.* The inequality $v_\pi(s, \tau') > v_\pi(s, \tau)$ immediately follows from Theorem 5. The inequality $v_\pi(s, \tau'') > v_\pi(s, \tau')$ follows from the same logic as the proof of Theorem 5 except applied to improving the policy in $(s, \tau')$. □

If each step of thinking in $s$ improves the policy, $\pi(s, \cdot)$, when should thinking terminate? From the proof of Theorem 5, we can see that thinking will terminate when the increase in value from thinking another step no longer compensates for the discounting of value caused by waiting a step to begin taking environmental actions. In this way, thought MDPs directly tie the "when to think" decision to the objective of reward maximization.

## 5 Language Generation as a Thought MDP

Our work takes inspiration from recent work on large reasoning models (LRMs) that "think" by generating additional text that is not part of the final answer but that somehow serves to improve the final answer. In this section, we review two approaches to imbue LLMs with reasoning capabilities and describe how they can be viewed as instantiating thought MDPs. We then show that forcing the LLM to reason (in a manner similar to zero-shot chain-of-thought [Kojima et al., 2023]) increases the expected return from a given state. Thus, Theorem 5 predicts that model-free RL applied to this thought MDP would lead to thinking, which is in fact what recent work has found.

We first describe language generation as an MDP and then describe the thought states and actions of LLMs. In language generation, each episode begins with a textual prompt, $x$ and the agent's actions are possible tokens from a fixed vocabulary. Let $y_t$ be the agent's output at time $t$. The state at time $t$ is defined as the prompt concatenated with the agent's outputs up to time $t$, $s_t = (x, y_{1:t})$. Rewards for RL applied to language generation have been defined in different ways. Much of the recent work on reasoning models focuses on math and coding problems, which have verifiable solutions. Thus, the reward is a terminal reward of 1 if a correct solution can be parsed and verified from $y_{1:t}$.

**Reasoning with Language as a Thought MDP** The main difficulty in mapping language generation to a thought MDP is that thought states are intertwined with environment states in the typical MDP formulation for LLMs. Similarly, thought actions are simply from the same space as environment actions. To distinguish these components, we redefine the environment state as just the query and tokens that function as part of the query response. Next, we define environment actions as just outputs that will affect how the overall response is evaluated and thought actions as the outputs that will not. Thus, the determination of whether one output is considered an environment or a thought action will depend upon the current context up to that decision. Finally, the thought state at time $t$ consists of the already produced tokens in $y_{1:t}$ that will not affect how the response is evaluated.

Different approaches to inducing reasoning in LLMs use different schemes for determining which tokens are part of the final response and which only serve to improve the final response. For example, Guo et al. [2025] augment the output vocabulary with two special functional tokens, `<think>` and `</think>`. In addition to the sparse verifier reward, Guo et al. train DeepSeek-R1 with reward shaping to encourage valid thinking blocks in which `<think>` is followed by `</think>`. Output text in thinking blocks is not part of the final response that is passed to a verifier to determine reward and would thus constitute the thought state. Zero-shot prompting is another approach to encourage thinking-like behavior by appending "Let's think step-by-step" to the query [Kojima et al., 2023]. In this approach, an answer parser is used to separate the response (environment actions) from the additional prompt and subsequent reasoning tokens (thought actions). The outputs that are discarded by the parser would constitute the thought actions.

| Model | No Thinking (%) | Thinking (%) |
|---|---|---|
| Qwen2.5-1.5B-Instruct | $7.20 \pm 0.82$ | $71.20 \pm 2.80$ |
| Qwen2.5-3B-Instruct | $6.80 \pm 0.80$ | $36.50 \pm 1.52$ |
| Qwen2.5-7B-Instruct | $5.06 \pm 3.10$ | $96.10 \pm 2.98$ |
| Qwen2.5-14B-Instruct | $0.90 \pm 0.33$ | $95.20 \pm 1.33$ |
| Tulu-2-7b | $0.30 \pm 0.33$ | $28.50 \pm 2.80$ |
| Tulu-2-13b | $0.60 \pm 0.47$ | $49.00 \pm 3.10$ |
| Llama-2-7b | $0.00 \pm 0.00$ | $48.80 \pm 3.10$ |
| Llama-2-13b | $0.20 \pm 0.27$ | $60.70 \pm 3.03$ |
| Gemma-3-1b-it | $0.50 \pm 0.43$ | $41.50 \pm 3.06$ |
| Gemma-3-4b-it | $4.90 \pm 1.33$ | $91.50 \pm 1.72$ |
| Mistral-7B-Instruct-v0.3 | $1.50 \pm 0.74$ | $85.20 \pm 2.20$ |

Table 1: Response accuracy $\pm$ 95% confidence interval when using thinking vs no thinking.

**Do thought actions improve expected return in LLMs?**   Recent work from Guo et al. [2025] has shown that thinking-like behavior can emerge from RL. Based on our theory, we hypothesize that a pre-condition for this result is that thought actions increase $v_\pi(s, \tau)$ by changing the thought state $\tau$. Because we lack a value function for each LLM, we instead approximate $v_\pi(s, \tau)$ with the Monte Carlo return or, equivalently, the accuracy of the LLM's response. Our hypothesis is that forcing the LLM to think will increase this value. Note that we use the Monte Carlo return as opposed to just looking at the immediate probability of the correct response (or its perplexity) to allow for the possibility that the LLM will generate its own reasoning and then answer correctly. To test our hypothesis, we take different pre-trained LLMs and apply them to add series of five four-digit numbers. We use models of various sizes from the following model familes: Qwen-2.5 [Yang et al., 2024, Qwen Team, 2024], Tulu-2 [Ivison et al., 2023], LLama-2 [Touvron et al., 2023], Gemma-3 [Gemma Team, 2025], and Mistral [Jiang et al., 2023]. We test two conditions with each model: "No Thinking" and "Thinking." Under the "No Thinking" condition, the prompt is "Compute the sum: [a] + [b] + [c] + [d] + [e] = " where [a], [b], [c], [d], and [e] correspond to four-digit integers. Under the "Thinking" condition, we append "[a + b] + [c] + [d] + [e] = [a + b + c] + [d] + [e] = [a + b + c + d] + [e] = " to this prompt, where [a + b] and [a + b + c] denote the partial sums a+b and a+b+c. In essence, we force the model to first think under the "Thinking" condition. We constuct 1000 "Thinking" and 1000 "No Thinking" prompts like these using 1000 different sequences of four 4-digit integers, each generated uniformly at random (see Appendix D for example prompts). Table 1 shows the average accuracy for each model. For all models, we see that appending the thinking tokens increases accuracy, which corresponds to increasing $v_\pi(s, \tau)$ by changing $\tau$. While we do not further apply model-free RL to try and learn to think in this way, our theory and these results predict that these models are primed for thinking to further emerge as a strategy.

## 6   A Non-Language Thought MDP

Our work takes inspiration from work on LLMs but aspires to understand how RL can lead to thinking in domains beyond language generation, possibly including agents that think in sequences of images [Ghazanfari et al., 2025, Wiedemer et al., 2025] or more abstract spaces [Hao et al., 2024]. One of the most important open questions is where do thought MDPs and initial thought-conditioned policies come from outside of LLMs? In this section, we hypothesize about generalized ingredients for a setting where thinking emerges as a useful strategy for reward maximization. Specifically, we hypothesize that the key ingredients may be multi-task pre-training coupled with the agent having the ability to manipulate its own internal state to activate pre-trained abilities. LLMs fit this hypothesis: self-supervised pre-training enables LLM agents to generate responses to many different contexts and language actions enable RL-finetuning to manipulate the input context to activate these existing capabilities when doing so will produce reward. To test this hypothesis, we set out to construct a non-language toy domain with these characteristics and to see if thinking could enable the agent to reach a higher return compared to non-thinking agents. Code for these experiments is available at https://github.com/prediction-action-lab/thinking-as-control.

We design a 5x5 gridworld where the agent can move in the cardinal directions or output one of three special actions, 'A', 'B', or 'C' that have no direct effect on the environment. Each special action corresponds to a possible task in the gridworld: 'A' corresponds to navigating to the lower

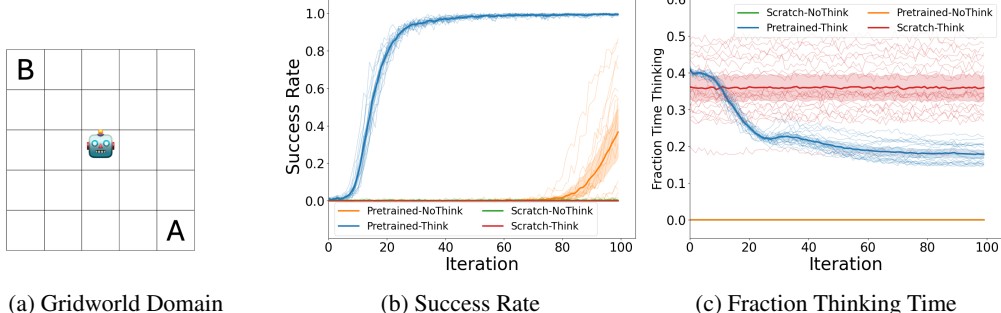

(a) Gridworld Domain      (b) Success Rate      (c) Fraction Thinking Time

Figure 2: Mean learning curves for the four agents we train in the gridworld environment. The vertical axis gives the success rate for first navigating to the bottom right and then the top left corner. The horizontal axis is the iteration of policy improvement (the agent collects 200 episodes at each iteration). We run 20 trials for each learning agent and shading indicates a 95% bootstrap confidence interval. Light lines show individual training runs.

right corner, 'B' corresponds to the upper left corner, and 'C' corresponds to going to the lower right corner and then to the upper left corner. Note that these special actions do not explicitly have these effects in the domain but will become associated with these behaviors through pre-training.[2] At the start of each episode, the agent receives one of these special actions as part of its observation.

We pre-train an initial policy through behavior cloning on an agent that "plays" in the gridworld. Specifically, at each time-step, this agent samples a new special action with probability 0.2 and subsequently acts optimally for the corresponding task until it succeeds or samples a new task (a new task is also sampled upon sub-task completion). During pre-training we only consider special actions 'A' and 'B' so that the agent has no experience with 'C'. The result of this procedure is a policy that wanders back and forth between the upper left and lower right corner while randomly changing its mind about where it is going. For the policy, we use a causal transformer [Vaswani et al., 2017] that conditions the probability of its action choice on the full episode history up to time $t$. As actions are included in this history, the special actions can manipulate the internal state of the agent even though they do not affect the environment state.

After pre-training, we use vanilla REINFORCE [Williams, 1992] and train the agent to complete the task corresponding to 'C'. The reward received is 1 when the agent is successful and 0 otherwise. Our hypothesis is that the ability to output 'A' or 'B' will enable a learning agent using the pre-trained transformer to activate skills learned during pre-training to complete the more challenging task. We test four agents: Pretrained-Think, Pretrained-NoThink, Scratch-Think, and Scratch-NoThink. The NoThink agents have the special actions masked and the Scratch agents are trained from scratch rather than initialized from the pre-trained model.

In Figure 2 we see that Pretrained-Think learns significantly faster than the other agents. The sparse reward makes learning from scratch impossible with or without thinking. Pretrained-NoThink also begins to learn the task, though it takes many more iterations to do so. This result was initially surprising to us and prompted further investigation on why pre-training helped even without thinking. The apparent reason is that sequences of environment actions can also serve to trigger the pre-trained policy's sub-task behavior in a way similar to thought actions. For example, if the agent moves down 2-3 times in a row then the agent will tend to continue to goal 'A' at the bottom right. Nevertheless, it is more difficult to discover such sequences compared to taking a single thought action.

We also investigate if thinking is actually what is learned by Pretrained-Think. Figure 2 shows that the agent initially takes a thought action about 30% of the time and this fraction falls as it learns and then stabilizes at about 15%. Optimal episodes are 14 steps long and the optimal amount of thinking – barring the agent figuring out the true optimal policy – is 2 times which is approximately 15% of all time-steps at convergence. We observe a trained agent and confirm that it first outputs 'A,' navigates to the bottom right, outputs 'B,' and then navigates to the top left.

Finally, we note that thought actions do not go away as the policy converges – in contrast to Proposition 4 which states that the optimal policy only takes environment actions. We suspect that

---

[2]In Appendix E, we also include results for the case where tasks 'A' and 'B' are slightly different from the sub-tasks needed to complete task 'C.'

this finding is due to sampling error that can cause poor convergence of on-policy policy gradient methods learning from finite samples [Corrado and Hanna, 2023]. As the agent discovers the strategy to first think and then act, it puts increasingly less probability on immediately taking the optimal action. Consequently, it never accidently discovers the benefit of taking immediate action to reach the goal a step or two earlier.

# 7 Limitations for Future Work

In this section, we discuss possible extensions for research in thought MDPs. Due to space constraints, we only describe a few directions while briefly noting that further research should consider extensions to partially observable worlds and connections to natural intelligence and the options framework.

**Where do thought MDPs come from?** This work studied the question of why thinking actions could be useful to an RL agent even though they leave their environment state unchanged and produce no reward. We formalized this abstract notion of thinking with the thought MDP model, but left open the big question of how to define the thought states, actions, and thought dynamics in other RL problems where thinking may be useful. Language generation is one example, but it is an open question as to how thought MDPs might arise outside the language domain. There is also the question of where existing thought-conditioned policies come from, as our work showed their existence to be a key enabler of emergent thinking. In Section 6, we explored the hypothesis that multi-task pre-training could be a key ingredient but this experiment was only designed as a proof-of-concept. In the future, it would be interesting to extend this exploration to more complex and realistic domains such as robots using pre-trained vision-language-action models or visual reasoning tasks using video models [Wiedemer et al., 2025].

**Connecting to Models and Planning** System 2 processing is reminiscent of decision-time planning using a model of the environment state transition function [Anthony et al., 2017]. This work has presented a more abstract model of thinking where the agent simply learns to control an internal thought state. While distinct models, both decision-time planning and thinking in a thought MDP are related in that they amount to locally focused policy improvement [Sutton and Barto, 2018]. Furthermore, having thought states and actions that are somehow grounded in environment states and actions could present an opportunity for explaining how the agents' thought dynamics should be structured. A starting point for this future work could be the Thinker architecture [Chung et al., 2023], which learns a single policy both for planning in a model (including deciding when to reset the search) and acting in the real world.

**Thinking in Dynamic and Time-constrained Domains** This work has only considered the utility of thinking in relatively static domains where the environment state remains the same during thinking. In reality, the environment state may change due to influences other than the agent's actions, and the choice to stop and think must factor in how the environment might change while it does so. A natural extension would be to have thought states and environment states unfold in parallel to one another, with the agent thinking and acting at the same time.

**Agents with Bounded Capacity** Could the utility of thinking be enhanced under constraints? We take inspiration from the Big World Hypothesis [Javed and Sutton, 2024] which states that agents will always have less capacity than what is required to learn all possible tasks. Consequently, when an agent is faced with a new task it may have either never seen it or have forgotten how to do it. Thinking might allow rapid repurposing of an agent's present capabilities to learn quickly on a new task. The sub-policies that are more frequently used would be repeatedly reinforced whereas less frequently used sub-policies could be forgotten.

# 8 Conclusion

In this work, we investigated the question of when model-free RL will lead to "thinking" behavior. We introduced the thought MDP model and then used this model to show that thinking emerges as a strategy to manipulate an agent's internal state so as to improve an agent's ultimate choice of environment action. Consequently, the emergence of thinking depends upon a policy initialization that implicitly contains sub-policies that can be triggered by taking thinking actions. We then provided supporting evidence that step-by-step reasoning in LLMs functions similarly to the thought actions in this theoretical model. Finally, we developed a non-language domain in which thinking would emerge as a strategy for reward maximization and discussed the many exciting next steps for developing AI agents that learn to think.

## Acknowledgments

We thank Xiaojin Zhu and Adam Labiosa for much discussion and comments that improved the final version of this work. This work took place in the Prediction and Action Lab (PAL) at the University of Wisconsin – Madison. PAL research is supported by NSF (IIS-2410981), the Wisconsin Alumni Research Foundation, and Sandia National Labs through a University Partnership Award.

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

# A    Proofs of Theoretical Results

**Proposition 7.** *Suppose we have a goal thought MDP that has $\mathcal{T} = \{\tau_0, \tau_1\}$. Let the initial policy, $\pi$, be such that there is some $p_0$ such that $\Pr(s_l \in s_{\texttt{goal}}|s, a, \pi(\tau_0)) > p_0$ for all $(s, a) \in \mathcal{S} \times \mathcal{A}$ and some $p_1$ such that $\Pr(s_l \in s_{\texttt{goal}}|s, a, \pi(\tau_1)) > p_1$ for all $(s, a) \in \mathcal{S} \times \mathcal{A}$. Let $c_s$ be the thought action such that $\tau_1 = p_\mathcal{T}(s, \tau_0, c_s)$ and let $p_c$ lower bound the probability $\pi(c_s|s, \tau_0)$. If $p_c \cdot p_1 > p_0$, then thought actions reduce the upper bound on the effective horizon of the MDP.*

*Proof.* The proof follows from the fact that, under the assumed policy initialization, thought actions raise the lower bound on finding a goal. First, note that the effective horizon has a tighter upper bound if we always fix the thought state to be $\tau_1$ rather than $\tau_0$. Thus, there is just the question of getting the thought state to be $\tau_1$. We lower bound this probability with $p_c$ so that, if the agent is in $(s, \tau_0)$, it has at least a probability of getting to $\tau_1$ from which it has at least a probability $p_1$ of finding a goal. Thus, the lower bound on $\Pr(s_T \in \mathcal{S}_{\texttt{goal}}|s, \tau, a, \pi)$ will be at least $p_c \cdot p_1$ for both $\tau_0$ and $\tau_1$ which reduces the effective horizon if $p_c \cdot p_1 > p_0$. $\qquad\square$

# B    Extended Theoretical Analysis for Finite-Horizon Thought MDPs

In the main paper text we formalized thought MDPs under the discounted return objective that is common in RL research. In this appendix, we extend these results to the finite-horizon, non-discounted setting that is common in contemporary uses of RL to train reasoning behavior in LLMs.

## B.1    RL in Time-Homogenous Finite-Horizon Markov Decision Processes

In this section and the following, we use $[k]$ to denote the set of integers $\{1, ..., k\}$. Formally, a time-homogenous, fixed-horizon MDP is a tuple, $\langle \mathcal{S}, \mathcal{A}, l, p, r \rangle$, where $\mathcal{S}$ is the environment state space, $\mathcal{A}$ is the set of actions that the agent can take to influence the environment state, $l$ is the maximum length of an episode, $p : \mathcal{S} \times \mathcal{A} \to \Delta(\mathcal{S})$ is a stochastic state transition function, and $r : \mathcal{S} \times \mathcal{A} \to \mathbb{R}$ is a scalar-valued reward function. When the agent is in state $s$ at timestep $t$ and takes action $a$ then it receives $r(s, a)$ and transition to a new state $S' \sim p(\cdot|s, a)$ and $t$ is incremented by 1. The process repeats from $S'$ until either a terminal state, $s_\infty$, is reached or $t$ reaches $l$. The agent selects actions according to a policy, $\pi : \mathcal{S} \times [l] \to \Delta(\mathcal{A})$. The time-dependent value of using a policy from a particular state, $s$, is defined to be $v_\pi(s, t) := \mathbf{E}_\pi[\sum_{i=t}^{l-1} r(s_i, a_i)|s_t = s, a_i \sim \pi(\cdot|s_i, i)]$. In finite-horizon RL, the agent's objective is to find a policy that maximizes $v_\pi(s, t)$ in all states and for all time-steps though, in practice, we could just focuse on maximizing $v_\pi(s, 0)$ for all possible initial states.

## B.2    Time-Homogenous Finite-Horizon Thought MDPs

We formally define a *finite-horizon thought MDP* as the tuple $\langle \mathcal{S}, \mathcal{A}, l, p, r, \mathcal{T}, \mathcal{C}, p_T \rangle$, where $\mathcal{S}, \mathcal{A}, p, r, l$ are defined as they are for standard finite-horizon MDPs. We add $\mathcal{T}$ as the set of *thought states*, $\mathcal{C}$ as the set of *thought actions*, and $p_T : \mathcal{S} \times \mathcal{T} \times \mathcal{C} \to \Delta(\mathcal{T})$ as the *thought transition function*. Recalling our informal definition of thinking from the main text, we emphasize that thought states and actions do not affect environment state transitions and rewards. However, they do affect the transition of time as we assume that the episode time-step will still be incremented when the agent takes a thought action.

We define policies for thought MDPs as in the main text except now with a time dependency: $\pi : \mathcal{S} \times \mathcal{T} \times [l] \to \Delta(\mathcal{A} \cup \mathcal{C})$. In this section, we will use the notation $\pi(\tau)$ to refer to the agent's state- and time-dependent policy with the thought state fixed at $\tau$.

In a thought MDP, interaction proceeds as follows. Episodes begin in timestep $t = 0$, initial environment state, $s_0$, and thought state, $\tau_0$, and the agent chooses either an environment action or thought action according to its policy. In either case, we increment $t$ to $t = 1$. If $a_0 \in \mathcal{A}$ then the environment state, $s_1$, at the next time-step is sampled from $p(s_0, a_0)$, the thought state $\tau_1$ keeps the value of $\tau_0$, and the agent receives reward $r_0 := r(s_0, a_0)$. Conversely, if $a_0 \in \mathcal{C}$, then $s_1$ keeps the value of $s_0$, $\tau_1$ is sampled from $p_T(s_0, \tau_0, a_0)$, and the agent receives $r_0 = 0$. Then the action-selection process repeats until either the agent reaches $s_\infty$ or $t = l$. For thought MDPs, we define the value function to be $v_\pi(s, \tau, t) := \mathbf{E}_\pi[\sum_{i=t}^{l-1} r(s_i, x_i)|s_t = s, \tau_t = \tau]$, with $r(s_i, x_i) = 0$ if $x_i \in \mathcal{C}$.

We next adapt the key assumptions of the main paper to the finite-horizon setting. First, we assume that thought state transitions are deterministic in order to simplify our analysis and because it is also the case for LLM agents.

**Assumption 9** (Deterministic Thought Transitions). $\forall s \in \mathcal{S}, \tau \in \mathcal{T}, c \in \mathcal{C}, p_{\mathcal{T}}(\tau'|s, \tau, c) = 1$ for one and only one $\tau' \in \mathcal{T}$. We will write $p_{\mathcal{T}}(s, \tau, c) \in \mathcal{T}$ to denote the thought state that results from taking $c$ in $(s, \tau)$.

Second, we assume that all rewards are non-negative as otherwise thinking could emerge as a strategy solely for the purpose of putting off receiving a negative reward, i.e., if all rewards are negative then the agent will be incentivized to just keep taking thought actions rather than environment actions.

**Assumption 10** (Non-negative Rewards). $\forall s \in \mathcal{S}, a \in \mathcal{A}, r(s, a) \geq 0$.

**Modeling of Time**   As in the discounted case, the fixed-horizon case raises questions about the passage of time. Just as in the main paper we assumed that discounting is applied equally between thinking and non-thinking time-steps, here we assume that both thought and environment actions take a single time-step. This assumption discourages thought actions as they cost the agent a chance to directly or indirectly obtain reward through an environment action. Potentially, we could consider varying times for thought and environment actions, however, this raises additional complexity that we will not consider further here. As before, the proposed model assumes that the environment state remains constant while the agent takes thought actions except for the passage of time. Again, for simplicity, we will focus on the static environment case, but note this issue as another interesting direction to refine the model. Finally, we only consider time-homogenous finite-horizon thought MDPs. In time-inhomogenous MDPs, the state transition probabilities and reward can differ across time-steps. In a time-inhomogenous finite-horizon thought MDP, thought actions might subtly change the state of the environment for the agent by allowing it to wait for more favorable state transition probabilities. A possible resolution would be to not advance the time-step when the agent selects a thought action. However, that resolution could break the finite-horizon assumption. For simplicity, we will only consider the time-homogenous finite-horizon setting.

**Optimality**   Finite-horizon thought MDPs are MDPs with state space $\mathcal{S} \times \mathcal{T} \times [l]$ and action space $\mathcal{A} \cup \mathcal{C}$ and consequently there will always be at least one deterministic optimal policy [Sutton and Barto, 2018]. Furthermore, we can show that this optimal policy will never take a thought action if we assume that environment actions are preferred to thought actions when the expected return is otherwise equal. The tie-breaking assumption means that thought actions are never strictly preferred to environment actions, though there may be situations when either type of action is equally preferred. Equal preference results in states from which it is impossible to obtain more future reward.

**Proposition 11.** *Assume that ties (w.r.t. expected reward-to-go) for the optimal action are broken in favor of environment actions. Then, any optimal policy, $\pi^\star$, for a thought MDP does not take thought actions: $\pi^\star(s, \tau, t) \in \mathcal{A}, \forall s \in \mathcal{S}, \tau \in \mathcal{T}, t \in [l]$.*

*Proof.* The proof is by induction. First, as the base case, consider the final time-step $l - 1$. Under the optimal policy, the agent receives $r(s, \pi^*(s, \tau, l-1))$ in $(s, \tau)$ and then interaction terminates. Since $r(s, \pi^*(s, \tau, l-1)) = 0$ if $\pi^*(\tau, \tau, l-1) \in \mathcal{C}$ and $r(s, \pi^*(s, \tau, l-1)) \geq 0$ if $\pi^*(\tau, \tau, l-1) \in \mathcal{A}$ then a thought action will never be strictly preferred to an environment action. Under the assumption that ties are broken in favor of environment actions, the optimal policy will select an environment action. Now, assume for arbitrary non-initial time-step $t > 0$ that the optimal policy only takes environment actions for all $(s, \tau)$ and for all subsequent time-steps $t' \geq t$. We need to show that $\pi^*(s, \tau, t-1) \in \mathcal{A}$. We will prove this step by contradiction. Suppose that $\pi^*(s, \tau, t-1) \in \mathcal{C}$. Then, after the agent takes $\pi^*(s, \tau, t-1)$, it transitions to $(s, \tau')$ and will then only take environment actions until termination (by our inductive hypothesis). However, if the agent changed $\pi^*(s, \tau, t-1)$ to take $\pi^{(}(s, \tau', t)$ then it would get the same expected total reward over the next $l - t$ time-steps and end up in some state, $s_{l-1}$ at time-step $l - 1$ from which it would have one more chance to take an environment action and gain non-zero reward. Consequently, changing $\pi^*(s, \tau, t-1)$ to this action would produce at least as high a return, which implies that $\pi^*(s, \tau, t-1) \notin \mathcal{C}$ if $\pi^*$ is the optimal policy and ties are broken in favor of environment actions.

$\square$

## C Policy Initialization Determines Emergence of Thinking

Finally, we extend Theorem 5 to the finite-horizon setting.

**Theorem 12.** *Let $\pi$ be a policy in a thought MDP such that $\pi(s, \tau, t) \in \mathcal{A}$ for some environment state $s$, thought state $\tau$, and time-step, $t$. If the policy improvement step of policy iteration sets $\pi'(s, \tau, t) \leftarrow c$ for $c \in \mathcal{C}$, then $v_\pi(s, \tau', t+1) > v_\pi(s, \tau, t)$, where $\tau'$ is the thought state resulting from taking $c$ in $(s, \tau)$ at step $t$.*

*Proof.* Policy iteration sets $\pi'(s, \tau, t) \leftarrow \arg\max_{x \in \mathcal{A} \cup \mathcal{C}} q_\pi(s, \tau, t, x)$ where

$$q_\pi(s, \tau, t, x) \coloneqq \begin{cases} r(s, x) + \mathbf{E}_{s'}[v_\pi(s', \tau, t+1)] & \text{if } x \in \mathcal{A} \\ v_\pi(s, \tau', t+1) & \text{if } x \in \mathcal{C}. \end{cases}$$

Since the policy improvement step is selecting a thought action, we have that:

$$\forall a \in \mathcal{A}, q_\pi(s, \tau, t, a) < v_\pi(s, \tau', t+1).$$

Since $\pi(s, \tau, t)$ was in $\mathcal{A}$ before the update we have that:

$$v_\pi(s, \tau, t) = q_\pi(s, \tau, t, \pi(s, \tau, t)) < v_\pi(s, \tau', t+1).$$

Thus, a thought action is selected when the policy is such that first changing the thought state to $\tau'$ leads to greater expected return than taking any environment action immediately, even though changing the thought state decreases the number of remaining time-steps in which the agent can obtain reward. $\square$

## D Example Prompts for LLM Experiment

Table Appendix D provides an example of the prompts we use in our LLM thinking vs no-thinking experiments.

| Condition | Prompt |
|---|---|
| No Thinking | Compute the sum: 5709 + 2890 + 4937 + 6482 + 6850 = |
| Thinking | Compute the sum: 5709 + 2890 + 4937 + 6482 + 6850 = 
 8599 + 4937 + 6482 + 6850 = 13536 + 6482 + 6850 = 20018 + 6850 = |

Table 2: Example "Thinking" and "No Thinking" prompts. The green text indicates the "thinking tokens" appended to the original "No Thinking" prompt.

## E Extended Empirical Description of Gridworld Experiments

This appendix provides additional details on our toy domain experiments.

**Implementation Details** We implement the Gridworld domain, pre-training, and reinforcement learning set-up in Python, using Pytorch [Paszke et al., 2019] for neural networks and gradient optimization. All experiments are ran on a Macbook Air with an Apple M1 chip and 16GB of memory. For both model pre-training and RL with REINFORCE we use the Adam optimizer [Kingma and Ba, 2015] with learning rates 1e-4 and 1e-5, respectively. For REINFORCE, we do not use a value function baseline as we found it generally did not help because the sparseness of the reward led to poor value estimates that harmed early policy learning. For the "NoThink" methods, we mask out the special actions by adding a large negative value to the logits for those actions before passing them to the softmax distribution.

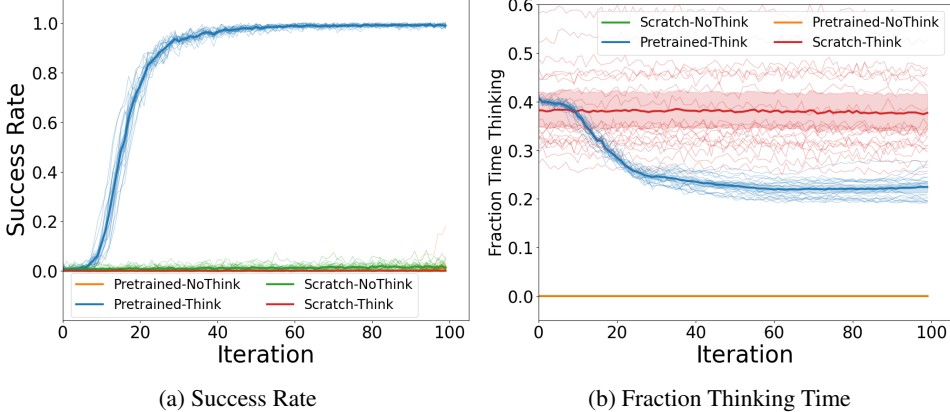

|                      |                      |
|:--------------------:|:--------------------:|
| (a) Success Rate     | (b) Fraction Thinking Time |

Figure 3: Mean learning curves for the four agents we train in the gridworld environment with misspecified sub-tasks. The vertical axis gives the success rate for first navigating to the bottom right and then the top left corner. The horizontal axis is the iteration of policy improvement (200 episodes are collected at each iteration). We run 20 trials for each learning agent and shading indicates a 95% bootstrap confidence interval. Light lines show individual training runs.

**Pre-trained Model Validation**    The objective of our experiment described in Section 6 was to create a setting where model-free RL (in this case REINFORCE) would lead to thinking as a strategy for reward maximization. A crucial piece of the set-up was to have a pre-trained model for which thought actions increase the agent's probability of solving the task. Recall that solving the task requires the agent to navigate to the bottom right corner and then to reach the top left corner. Pre-training is designed so that the agent outputting special action 'A' increases the probability of it then taking the actions that lead to the bottom right and similarly so that 'B' increases the probability of moving to the top left. We validate the pre-training procedure by taking pre-trained models and forcing their first action to be 'A' and their action upon reaching the bottom right for the first time to be 'B'. We find that the pre-training procedure leads to models where such prompting increases the probability of task success compared to simply rolling out the model. If we further prompt the model this way and also mask thought actions on every other step then the pre-trained model will solve the task on virtually every episode. This prompting and masking procedure confirmed that pre-training had primed the model for thinking to emerge as an effective strategy.

**Pre-training with Misspecified Sub-Tasks**    In the Gridworld domain, we pre-train the agent's policy so as to associate the thought actions 'A' and 'B' with two sub-tasks that will be needed on the final evaluation task. Naturally, one might wonder whether thinking will still emerge if the pre-trained sub-policies are misspecified for the final learning task. To test this, we pre-train a model using the exact procedure described above but change the evaluation domain so that the complex task has each sub-goal location shifted by one cell. Though a small change, it is a sufficient change so that the strategy discovered in our main task setting will no longer completely solve the task. With this set-up, we test whether the agent can still leverage thinking to solve the task, even though triggering the pre-trained behaviors is insufficient by itself. Figure 3 shows similar results as Figure 2 in which the pre-trained method that uses thinking actions remains the only method to fully solve the task. Interestingly, we no longer observe any runs of the pre-trained no-think agent successfully completing the task.

