# OpenReview forum: "When Can Model-Free Reinforcement Learning be Enough for Thinking?"
_NeurIPS.cc/2025/Conference — NeurIPS 2025 poster_

### Official Review · Reviewer_pJ6m · 2025-06-28

**Clarity:** 3
**Significance:** 3
**Originality:** 3
**Rating:** 4
**Confidence:** 4

**Summary:**

This paper developed a formal framework studying the role of thoughts in reward maximization. The paper proposes a model of "thought MDP", which augments the conventional MDP for RL modeling with thought states and actions. With this formulation, the paper shows that thought-augmented MDP improves policy and is more effective in maximizing rewards. The paper also shows that policy initialization is critical to the emergence of thoughts, and how the thought MDP helps language and non-language models to maximize rewards.

**Questions:**

1. While the paper provided analysis of how thinking improves policy, it remains unclear what conditions of the thinking steps need to satisfy so that policy can be improved and reward optimized. An agent's performance cannot be improved with arbitrary thought states and actions, it would be great to provide some theoretical characterization of the quality of thoughts.

2. In some sense the "thought MDP" can be regarded as introducing a latent variable to the MDP process. It would be great if the paper could provide some comparisons between the proposed framework differ and existing analysis on RL with latent variables, e.g.

"Stochastic Latent Actor-Critic: Deep Reinforcement Learning with a Latent Variable Model", NeurIPS 2020
"Latent Thought Models with Variational Bayes Inference-Time Computation", arXiv 2025
"Latent Variable Representation for Reinforcement Learning", ICLR 2023
"A Theoretical Understanding of Chain-of-Thought: Coherent Reasoning and Error-Aware Demonstration", arXiv, 2024

**Ethical Concerns:**

["NO or VERY MINOR ethics concerns only"]

**Final Justification:**

The development of a rigorous formalization mechanism for the analysis of COT and reward optimization deserves follow-up work and discussion. I think overall this is a well-written paper, although some of the author responses could be better incorporated in the submission for improved clarity.

**Limitations:**

Yes.

**Quality:**

3

**Strengths And Weaknesses:**

Strengths:
This paper provides a rigorous formal mechanism that elucidates the role of thoughts in reward optimization of RL, which offers a solid theoretical contribution to the understanding of how COT + RL improves agent model's performance.

Weakness:
Although the paper's title is "Is reward enough for thinking", this paper actually touches several sub-topics (how thinking helps reward optimization, condition for thought emergence, language/non-language thought MDP etc.). This divergence makes the central argument somewhat difficult to follow. A clearer high-level summary and more structured line of reasoning would strengthen the paper's coherence.

The paper could also have compared existing theoretical analysis on the role of COT and latent thinking steps.

---

> ### Author Rebuttal · Authors · 2025-07-30
>
> Thanks for your kind comments on the strength of our theoretical analysis!
>
> Weaknesses:
> 1. We agree the title could be improved and can easily address this for the camera-ready subject to PC Chair approval. A more appropriate title might be “When does model-free RL lead to thinking?” or something similar.
> 2. This is a good suggestion that we can easily add to the related work. In terms of understanding COT, the uniqueness of our work seems to be explaining the emergence of reasoning in RL terms. We’ll discuss other papers you suggest below.
>
> Questions:
> 1. This is a good point that we can clarify. We are not claiming that arbitrary thought states and actions will lead to better performance. The implication of Theorem 1 is that, if there is a thought action that leads to a thought state where the policy performs better than under the current state then that thought action will improve the agent’s performance.
> 2. Thanks for suggesting these references. We can add discussion about how existing work on COT and latent states relates to our work. Specifically:
>       - Prior work on COT seems to focus on why COT outputs are useful even prior to any RL training being performed. To the best of our knowledge, these works are not considering the emergence of thinking via RL. Our work answers the question of when COT outputs will be selected for by RL.
>      - In the RL literature, latent state variables are usually discussed in partially observable settings. In such settings, the agent can try to construct a state representation that still supports good decision-making. Alternatively, in deep RL, the latent state can be a representation of state that enables predicting state values using a simpler function (e.g., a representation that enables linear value prediction). In contrast, we are focusing on a fully observable environment and analyzing the tabular setting where we have appropriate representational power to learn the value function and policy. The thought states serve a novel purpose which is to give the agent a variable to manipulate to trigger different sub-policies within its policy function.
>
> Please let us know if we can provide further clarification!

---

> > ### Comment · Reviewer_pJ6m · 2025-08-04
> >
> > Thank you authors for your responses. For one, I do think a title like "When does model-free RL lead to thinking?" more accurately reflects the core argument of the paper. Overall, the authors have addressed my questions properly, and I consider this a good paper that merits discussion within the research community.

---

> > > ### Author Response · Authors · 2025-08-05
> > >
> > > Thank you for the feedback on the title and additional kind words. We're happy to hear that we addressed your questions!

---

### Official Review · Reviewer_RSQS · 2025-07-03

**Clarity:** 4
**Significance:** 2
**Originality:** 4
**Rating:** 4
**Confidence:** 3

**Summary:**

The author extends standard MDP by adding a thought action space, such that at each step the agent may either act in an environment ($A$) or take a zero-reward "think" step ($C$) that deterministically updates an internal thought state ($\tau$) while environment state (s) remains unchanged. Under these three assumptions:
- thought transitions are deterministic,
- all external rewards are non-negative
- for any state there exists a policy that can eventually reach a non-zero reward with finite number of steps,

the authors prove:
- a fully-optimal policy never thinks,
- during learning a thought step is selected exactly when its discounted value exceeds the value of acting immediately

Thus, thinking emerges only when it raises expected returns enough to offset the discount. Experiment using 4-digit addition shows thought actions indeed improve expected return in LLMs as perplexity significantly reduced.

**Questions:**

- Could authors provide a training recipe along with an experiment showing thinking emergence during training? Using the multi-digit addition with a small LLM should suffice.
- Many existing RL for LLM is implemented using token-level rewards where a single trajectory reward gets broadcasted to every position of the token. Does this work assume discount $\gamma$ apply at the whole generation level as well?
- What happens when there is no discount factor applied to thinking? $\gamma=1$

**Ethical Concerns:**

["NO or VERY MINOR ethics concerns only"]

**Final Justification:**

The author has addressed my concerns and answered my questions. Overall i think it's a good paper with insights that would be beneficial to share with the community.

**Limitations:**

Yes

**Quality:**

3

**Strengths And Weaknesses:**

Strengths:
- The formalization is clean and correctly derived. The paper gives a unified theory to explain conditions in which a bunch of test-time compute scaling strategies arise during RL training.
- The proofs clearly separate optimal behavior from learning time, suboptimal behavior and show how reflection can shorten the effective horizon in sparse-ward tasks. This is particular useful to explain recent effort in trying to make model think more efficiently by applying discount during training.

Weaknesses:
- The LLM-based experiments is in a trivial domain (4-digit addition), and it's hard to see how it generalizes to more complicated tasks.
- The paper does not provide a training recipe or concrete algorithm for training LLMs despite showing the value estimates as perplexity.

---

> ### Author Rebuttal · Authors · 2025-07-30
>
> Thanks for your kind comments on the formalization and theory. We're happy to hear that you lean toward accepting the paper and we hope that we can address some of the weaknesses in a camera-ready version.
>
> Weaknesses:
> 1. We chose addition for its simplicity and because it is straightforward to create correct reasoning chains for it (i.e., to create the "thinking" condition). We will think about potential other tasks to consider. If you have particular tasks in mind, we would welcome the suggestion!
> 2. Producing a new algorithm for LLMs wasn't one of the goals of our work. Our paper's main contribution is to develop the theory around when model-free RL will produce thought actions. The purpose of this is to provide a deeper understanding of this new type of thinking that recent work in training LLMs to reason has provided a proof-of-concept for.
>
> Questions:
> 1. We don't think our work immediately suggests new training recipes for LLMs. Rather we are trying to understand why current training recipes lead to thinking emerging.
> 2. In formal terms, many existing works optimize the undiscounted return over fixed-length episodes. In our theory we consider the discounted return over indefinite length episodes. We will add more discussion around this point to the camera-ready. In our response to reviewer zETt, we explain that the discounted return may be a more appropriate objective for developing a general theory of thinking agents and also discuss the relatively minor extension of our theory to the undiscounted, fixed-length episode case. Please let us know if this doesn't address your concern!
> 3. To sum-up our response to reviewer zETt for $\gamma=1$: having $\gamma=1$ requires that episodes can't have infinite length and so we also adopt the fixed-length episodic setting (this also matches LLM training set-ups where generation can't continue indefinitely). The simple extension of our theory is to make the value function time-dependent. Under this modification, the analysis is qualitatively the same and tells us that thinking will be selected for when an additional step of thinking improves the policy enough to offset losing a time-step to collect reward.
>      - If you mean what happens if we only discount non-thinking steps, the conclusion is that thinking will go on for as long as $v_\pi(s,\tau)$ keeps improving w.r.t. $\tau$.
>
> Please let us know if you'd like to discuss these questions further!

---

> > ### Comment · Reviewer_RSQS · 2025-08-07
> >
> > Thank you for your thorough responses. My questions are addressed and I don't have any more follow up. Overall i think it's a  good paper with insights that would be beneficial to share with the community.

---

> > > ### Author Response · Authors · 2025-08-07
> > >
> > > Thank you so much for the kind words!

---

### Official Review · Reviewer_vneH · 2025-07-03

**Clarity:** 2
**Significance:** 3
**Originality:** 3
**Rating:** 4
**Confidence:** 2

**Summary:**

This paper studies the emergence of thinking in Large Language Models (LLMs). Recent works have discovered that model-free reinforcement learning with special tokens can encourage LLMs to think (e.g., generating long chain-of-thought) and improve reasoning performance. To study this phenomenon, this paper models the reasoning process of LLMs as a Thought Markov Decision Process. Thus, reasoning is to learn the transition between the thought states. Furthermore, it develops a theoretical framework to show that initialization is important for the emergence of thinking. It further discusses the emergence of thinking in other domains.

**Questions:**

I am \textbf{not} a theoretical researcher. Please address the weaknesses part above.

**Ethical Concerns:**

["NO or VERY MINOR ethics concerns only"]

**Final Justification:**

After reading the reviewer's response and additional experiments, I think my concerns are addressed and I will keep the scores. The reason why I do not increase to 5 is due to the mismatch with my expertise. I do not work on theory, so I am not in a good position to evaluate the theoretical contribution of this manuscript. But the empirical finding of this manuscript aligns well with the literature.
So I will keep my original assessment.

**Limitations:**

Yes

**Quality:**

3

**Strengths And Weaknesses:**

Strength:

1. This paper studies how thinking emerges from LLMs, which is receiving increasing attention. Empirical findings reveal that model-free reinforcement learning can improve the reasoning ability of LLMs by teaching them to think. Although such a paradigm has achieved superior performance in mathematical reasoning and coding [1], the mechanism behind this emergence of thinking is unknown. Recent work [1] also questioned whether RL can improve the reasoning ability of LLMs. To the best of my knowledge, this paper is one of the earliest ones to study this problem from a theoretical perspective.

2. To study the emergence of thinking, this paper established a Thought MDP and theoretically analyzed the importance of policy initialization to thinking. This finding aligns well with recent empirical results, as shown in [2].

Weakness:

My major concerns are about the clarity and empirical evaluation. For example, what are the thought states and actions in the reasoning process? It is better to have some illustrative examples. Moreover, the evaluations in Section 5 are only conducted on small scale models (~1B). It would be better to evaluate on large-scale models.


[1]. Deepseek-r1: Incentivizing reasoning capability in LLMs via reinforcement learning.

[2]. Does reinforcement learning really incentivize reasoning capacity in LLMs beyond the base model?

---

> ### Author Rebuttal · Authors · 2025-07-30
>
> Thanks for your kind comments on our novelty and alignment with empirical findings. We're happy the theoretical contribution is appreciated.
>
> Weaknesses
> 1. As we aim to develop a domain-agnostic view of what it means for an RL agent to think, we intentionally define thought states and actions abstractly. We agree that an example would improve clarity and can provide one in the camera-ready version. Specifically, we can move up the discussion of how applying RL for LLM learning-to-reason is instantiating a thought MDP.
> 2. We can easily add evaluations with larger models to the camera ready. Since the submission, we have ran this experiment with 11 different models, including Qwen2.5-14B, Llama-2-13B, and Tulu-2-13B. The other models we will include range from 1-7B parameters. There is no qualitative difference between the new results and the ones with the smaller models. See below for discussion.
>
> We include additional evaluations with 11 different models ranging from 1.5B - 14B parameters in the table below. Since submission, we have made a minor change to our experimental setup: rather than using perplexity to indirectly infer changes to value of model responses, we instead directly approximate state values with the Monte Carlo return, which is equivalent to the accuracy of the LLM’s response. We observe no qualitative difference between the new results and the ones with the smaller models: appending thinking tokens to the input prompt increases the value of the response accuracy.
>
> 	| Model                     | No Thinking (%)       | Thinking (%)          |
> 	|--------------------------|-----------------------|------------------------|
> 	| Qwen2.5-1.5B-Instruct    | 7.20 ± 0.82           | 71.20 ± 2.80           |
> 	| Qwen2.5-3B-Instruct      | 6.80 ± 0.80           | 36.50 ± 1.52           |
> 	| Qwen2.5-7B-Instruct      | 5.06 ± 3.10           | 96.10 ± 2.98           |
> 	| Qwen2.5-14B-Instruct     | 0.90 ± 0.33           | 95.20 ± 1.33           |
> 	| Tulu-2-7b                | 0.30 ± 0.33           | 28.50 ± 2.80           |
> 	| Tulu-2-13b               | 0.60 ± 0.47           | 49.00 ± 3.10           |
> 	| Llama-2-7b               | 0.00 ± 0.00           | 48.80 ± 3.10           |
> 	| Llama-2-13b              | 0.20 ± 0.27           | 60.70 ± 3.03           |
> 	| Gemma-3-1b-it            | 0.50 ± 0.43           | 41.50 ± 3.06           |
> 	| Gemma-3-4b-it            | 4.90 ± 1.33           | 91.50 ± 1.72           |
> 	| Mistral-7B-Instruct-v0.3 | 1.50 ± 0.74           | 85.20 ± 2.20           |
>
> Table. Response accuracy with ± 95% confidence interval when using thinking vs no thinking.

---

> > ### Comment · Reviewer_vneH · 2025-08-04
> > **Thanks for the response! I will keep the score.**
> >
> > Thank you very much for the extensive evaluation of different model sizes. This effectively addresses my concerns. I will keep the original score.

---

> > > ### Author Response · Authors · 2025-08-05
> > >
> > > We're happy to hear that your concerns have been addressed! FYI, for some reason (and this might be an OpenReview issue), your score seems to have disappeared from the review.

---

> > > > ### Comment · Reviewer_vneH · 2025-08-07
> > > > **Score disappear**
> > > >
> > > > I guess it might be related to OpenReview. I did not change the initial score.

---

> > > > > ### Author Response · Authors · 2025-08-07
> > > > >
> > > > > Thanks for the reply. After posting this reply to you, the PC emailed to say that this is the intended behavior to hide ratings in some cases.

---

### Official Review · Reviewer_QBee · 2025-07-03

**Clarity:** 3
**Significance:** 4
**Originality:** 4
**Rating:** 5
**Confidence:** 4

**Summary:**

In this paper the authors explored why "thinking" (defined as agent actions that does not produce reward and does not modify the environment state) might emerge as part of model-free RL learning. The proposed a framework named Thinking MDP to ground their research. They provided theoretical results on 1) the benefit of thinking as a form of test-time policy improvement and 2) how policy initialization influences the whether thinking emerges.

The authors further provided toy example to demonstrate that 1) thinking could improve LLM's policy at test time, and therefore would be expected to emerge as a policy during RL, though the final optimal policy would not require thinking. And 2) some empirical evidence that multi-task pretraining is a core criteria for models where thinking could be useful (as a form of helping the policy navigate to learned sub-policies).

**Questions:**

1. In Section 6, why does the thinking action still occupies 15% of final policy if Proposition 4 suggests an optimal policy doesn't need thinking?

2. Suggestion: provide some empirical evidence on the emergence of thinking in LLMs as RL iteration goes on, similar to Figure 1 but in a real-life LLM setting.

3. Provide more ablations on the effects of multi-task pretraining, for example if the pretraining tasks are not strictly "subtasks" for the RL task, what would happen.

**Ethical Concerns:**

["NO or VERY MINOR ethics concerns only"]

**Final Justification:**

Thanks for addressing the comments. I will keep the original score.

**Limitations:**

Yes

**Paper Formatting Concerns:**

No major concerns

**Quality:**

3

**Strengths And Weaknesses:**

Strength:
- Well written paper.
- Provided strong theoretical results on how thinking can be a viewed as a form of test-time policy improvement given suboptimal policy / value functions.
- Provided interesting demonstration on their theoretical results.

Weakness:
- The paper stopped short of showing that "some LLM can benefit from thinking" necessarily implies the "some LLM will get thinking ability through model-free RL". The authors argue that if an optimal policy can be learned then thinking is necessary, but some additional research in realistic settings (RL on pretrained LLM) that thinking emerges and then goes away.
- The paper did not provide much grounding on their claim that multi-task pretraining is key criteria for thinking. Do the authors mean that thinking is essentially a way to generalize from learned sub-tasks during multi-task pretraining? Some more exploration on this could be meaningful.
- The paper hinted that thinking shares certain similarity with MCTS. This could be explored more in settings such as teaching a LLM to play a game such as Chess.
- Some minor grammar mistakes.

---

> ### Author Rebuttal · Authors · 2025-07-30
>
> Thank you for your kind remarks—we’re glad you found the paper interesting!
>
> Weaknesses:
> - We agree that this is the correct phrasing of the claim we make—these LLMs can benefit from thinking. And you bring up an interesting point about would thinking emerge and then go away. While our theory focuses on exact policy iteration, we conjecture that thinking will not necessarily go away in practical policy gradient training and this connects to your later question about why in Section 6 that thought actions still are taken 15% of the time. Unlike exact policy iteration, policy gradient algorithms require sampling from the action space. Consequently, once the policy has put too much probability on the thinking actions, it is unlikely to sample the true optimal environment action, discover that such an action enables solving the task slightly quicker, and consequently converges sub-optimally.
> - Regarding multi-task pre-training, our intention is to show that thinking is a way to trigger behaviors contained within the policy via manipulation of the agent’s internal thought state. If the policy doesn’t contain any such useful behaviors, there is nothing to trigger and so thinking is not useful. Pre-training appears to be the main possibility for setting up a policy to have such behaviors—though of course there may be other ways. We do not claim pre-training is necessary but it may be sufficient.
> - We agree that exploring connections between MCTS (a more classic System 2 approach) and the thought MDP model is a fascinating direction for future work. This is on our research agenda and the idea to look at an LLM engaged in a task that requires planning would be cool to explore.
> - We will carefully proofread the paper before the camera ready deadline.
>
> Questions:
> 1. See our response to the first weakness. Proposition 4 concerns optimal behavior that stochastic learning algorithms may not necessarily converge to. In principle, if we had a tabular policy representation and a sufficiently large batch size for estimating the policy gradient then we would expect thinking to go away. However, this may not happen when using transformer policies and finite batch sizes. See [1] for more discussion on how PG methods can converge sub-optimally due to sampling variation. We will add discussion of this important point in the paper.
> 2. Thanks, we agree this is a valuable addition and will look into adding it. We do note that this evidence is already present in the literature for LLMs in general, though not necessarily for the specific ones we consider.
> 3. This is an interesting suggestion and we can easily do this experiment and include in the camera-ready. The natural ablation would be to make the pretraining tasks to move to different, non-corner locations or to the corners that aren’t sub-goals in the eventual task. We expect thinking to still be useful but overall less effective in these cases.
>
> [1] On-Policy Policy Gradient Reinforcement Learning Without On-Policy Sampling. Corrado and Hanna. Arxiv 2024.

---

### Official Review · Reviewer_zETt · 2025-07-06

**Clarity:** 3
**Significance:** 2
**Originality:** 3
**Rating:** 3
**Confidence:** 4

**Summary:**

This work studies the question: "when is it possible, from an RL theoretic point of view, to encourage 'thinking' steps by just pure RL?" They formally define a construct called a "Thinking MDP" which is an MDP which incorporates "thinking states" and "thinking actions". The reward is discounted for each thinking or acting step taken, in accordance with standard MDPs, and is assumed to be non-negative. This formulation leads to several mathematical propositions: an optimal policy for a thought MDP never takes any thinking steps, but policy iteration algorithms may sometimes empower the model to take thinking steps if the expected reward is better under the current policy; and thinking MDPs may also have a smaller effective horizon (i.e., are "less complex") than their ordinary counterparts, under some cases. The authors attempt to validate their theory using some controlled experiments in LLMs and outside of LLM context using purely synthetic settings.

**Questions:**

- How resilient are the theoretical results to the specific assumptions made? Is it possible to adapt the results to other assumptions?
- How can we formalize the state/actions in terms of the LLM token generation process?
- Can a modified RL procedure which uses the assumptions made in the paper actually do RL training of a fine tuned model? (I.e., instead of making theory fit the practice, you can instead introduce a new practical method which fits the theory and show that it does reasonably well.)

**Ethical Concerns:**

["NO or VERY MINOR ethics concerns only"]

**Final Justification:**

I have read the rebuttal. I have improved the score given the authors' reply and other reviewers' opinions. But I still believe the paper is below the bar for acceptance.

**Limitations:**

Yes

**Quality:**

2

**Strengths And Weaknesses:**

Strengths:
- The Thought MDP formulation is interesting, and (potentially one of) the first attempts to connect thinking/reasoning in LLMs to RL theory.
- The mathematical consequences of the formulation seem solid and interesting, albeit very connected with the specific assumptions made, which may not be realistic.
- The synthetic experiments in non-LLM RL environments demonstrate the possible generalization of reasoning RL methodology outside of LLMs.
Weaknesses:
- The thought MDP formulation is not fully connected to LLMs. In reasoning LLM training, my understanding is that each action or thought is a full "turn", not just a single token, and this affects the reward formulation. Often, there is no discount factor, and in any case the discount factor often does not apply to thinking steps. In this sense, the reward formulation of the thinking MDP in this paper seems like it does not apply to the practice of LLMs.
- The synthetic experiment with LLMs is also similarly heuristic, and in particular, does not actually apply RL to the model, so it is hard to show how the results in this experiment corroborate the theoretical results.
- The theoretical assumptions, which seem not very applicable to practice, seem to underpin all the theoretical results, so these may not apply either or be slightly misleading when applied to practice.

---

> ### Author Rebuttal · Authors · 2025-07-30
>
> We’re happy to hear you found the formulation and its consequences interesting! Your comments were very though-provoking for us and we will add more justification around our assumptions based on your comments.
>
> You’re correct that the formulation is not fully connected to current LLM practice—we see that as a strength rather than a weakness for theory work. As noted in the global comment above, our objective is foremost to develop a general model of this new way to develop “thinking” in AI agents. While LLMs are a proof-of-concept that this type of model can exist and be practically impactful, we see this as just the beginning for this general methodology. After all, humans don’t just think in language but also in images and more abstract thought spaces.
>
> That said, our theoretical assumptions are not so disconnected from RL for thinking LLMs in practice as we read the literature.
> - As we understand it, when training LLMs to reason, each action is a single token. See [1], Section 3.1 for the MDP formulation that we understand to be widely used. Sometimes papers describing RL for LLMs describe the LLM as a policy over sequences. Since the environment is deterministic and the model auto-regressive, this formulation is valid, but when it comes to implementation, tokens are the actions actually produced by the transformer model. Of course, you are right that a single token may have limited effect by itself.
> - You’re correct that discounting is not standard in current LLM training (as far as we can tell). However, we would argue that some form of discounting is ultimately essential in any complete model of an AI agent, and hence we include it in our theoretical model. Otherwise, the agent may simply think forever without consequence. Perhaps this is part of why some reasoning models (e.g., DeepSeek-R1) respond to simple math problems with long-winded thinking traces [2,3].
> - We also note that we could potentially deprecate the use of discounting in our theory by saying there is a maximum episode length and additionally conditioning the value-function on the episode time-step. The lesson from the theory would still be qualitatively the same: thinking is selected for when the amount of policy improvement that it brings compensates for having one fewer time-step to take actions that could directly produce reward.
>
> The LLM experiments are intended to show that the right sequence of “thoughts” will increase the expected return of the agent under the current policy (i.e., base LLM). Showing this confirms that this sequence of actions would be preferred by a step of policy improvement (as we look at sequences, the result is more related to Corollary 1). We will clarify the extent to which the empirical results corroborate the theory—and discuss the limitations of what they show.
>
> Regarding our theoretical assumptions, are there other assumptions beyond the disconnects you mentioned above that you would like to discuss? Here we discuss each key assumption made in the paper:
> 1. Assumption 1 (deterministic thought transitions): We initially defined Thought MDPs as having stochastic thought transitions for generality. This assumption is straightforward to relax but we didn’t do it for ease of exposition. Deterministic thought transitions are also the case with LLMs. We can add the generalized results to the appendix in the camera ready.
> 2. Assumption 2 (non-negative rewards): This assumption matches current practice in applying RL to verifiable reward signals (i.e., 0/1 rewards). The consequence of not having this assumption is that, in general, the agent might learn to think forever to avoid receiving negative reward.
> 3. Assumption 3 (reachable positive reward): This assumption simply states that the agent always could reach some positive reward in finite time from any state. This will generally hold for LLM agents and is also a mild assumption that basically says the MDP is interesting in the sense that the agent can possibly produce reward.
>
> Questions:
> 1. Above we have discussed ways to adapt the key assumptions. If we’re missing some assumptions that you would like to discuss, please let us know.
> 2. See discussion above. Our understanding is that works using RL on LLMs are treating tokens as actions since tokens are the outputs of the underlying auto-regressive transformer model.
> 3. Based on our discussion above, we’re not sure what this method would look like as we believe many of our assumptions are relevant to LLM training. We want to emphasize that developing a new procedure for training LLMs was NOT a goal of our work.
>
> We hope our response addresses your concerns and we are happy to discuss any of these points further!
>
> [1] DPO meets PPO: Reinforced Roken Optimization for RLHF
>
> [2] Stop Anthropomorphizing Intermediate Tokens as Reasoning/Thinking Traces!
>
> [3] RL in Name Only? Analyzing the Structural Assumptions in RL post-training for LLMs

---

> > ### Author Response · Authors · 2025-08-07
> >
> > Dear reviewer zETt,
> >
> > Thanks for your initial review. As the discussion period is closed to finish, we would appreciate the opportunity to engage with you on the issues you raised and to be able to understand if we have addressed your concerns about the assumptions we make.
> >
> > Thanks for your time!
> >
> > Best,
> > The authors

---

### Note · Authors · 2025-08-13

Dear AC and Review Team,

Once again, thank you for reviewing our paper, many kind words, and suggestions for improvement.

We recently noticed that our response to Reviewer zETt referenced a "global comment" that we wrote but then realized could not post during the response period. To make sure our response is complete, we include that here as a concluding remark:

>We thank all the reviewers for their comments and are happy for the overall positive response to our work. We particularly appreciated that the reviewers noted:
> 1. We make arguably one of the first attempts to connect thinking in LLMs to RL theory (zETt,vneH,)
>2. We provide a rigorous interpretation of the mechanism by which thinking emerges (QBee, pJ6M)
>3. We provide the first instantiation of how thinking can emerge in a non-language domain (zETt, vneH)

>The main concern noted seems to be about places where our analysis is disconnected from current practice in using RL for reasoning in LLMs. We’ll address specifics of this concern below (primarily for reviewer zETt). Here, we note that our work aimed to go beyond a theoretical foundation for just the current practice for developing LLM reasoning with RL. Rather, we want to provide a theoretical foundation for an entirely new type of thinking behavior that is emerging in the AI literature. LLMs are an important proof-of-concept for the emergence of this new type of learning-to-think behavior and current practice shows its possibility in frontier systems. Consequently, we did anchor our theoretical model on them in many respects. But we also chose not to limit our model to current RL-for-reasoning LLM practice (which is fast evolving in any case) so that it might capture aspects of learning to think that we conjecture to be fundamental.


Thank you again for your kind words and time on this!

Best,

The authors

---

### Decision · Program_Chairs · 2025-09-17

**Decision:**

Accept (poster)

**Comment:**

This paper presents a novel and timely theoretical exploration into the phenomenon of "thinking" as it emerges from model-free reinforcement learning, particularly in the context of large language models. The authors introduce a formal framework, the "thought Markov Decision Process" (thought MDP), to analyze the conditions under which an agent would choose to perform internal, non-reward-generating "thought actions." Through this lens, they elegantly prove that thinking can be understood as a form of on-the-fly policy improvement. While an optimal policy would not need to think, the paper shows that during learning, thinking can emerge as a valuable strategy if the expected future reward outweighs the immediate cost of computation, with policy initialization playing a critical role. The theoretical claims are supported by illustrative experiments in both LLM and non-language domains, providing a compelling, domain-independent perspective on this important emergent capability.

I recommend acceptance for this paper. It was well-received by 4 of the 5 reviewers, who lauded it as a pioneering effort to formally connect the empirical success of chain-of-thought-style reasoning with foundational RL theory. The reviewers found the thought MDP formulation to be clean, insightful, and a significant contribution. While valid concerns were raised regarding the current gap between the paper's theoretical assumptions and the complex realities of training frontier LLMs, the authors' rebuttal has persuasively clarified their goal of building a more fundamental, lasting theory rather than just modeling today's evolving practices. It is also worth noting that the single outlier reviewer, despite a lack of engagement during the discussion period, raised their score and acknowledged the paper's strengths. I am confident that the authors have gracefully taken the constructive feedback to heart and that the final camera-ready version, incorporating the reviewers' excellent suggestions, will be a valuable contribution to the NeurIPS community.